# The Regularizing Effect of Different Output Layer Designs in Deep Neural Networks

## Abstract

Deep neural networks are prone to overfitting, especially on small datasets. Common regularizers such as dropout or dropconnect reduce overfitting, but are complex and prone to hyperparameter choices, thus prolonging development cycles in practice. In this paper, we propose simple but effective design changes to the output layer - namely randomization, sparsity, activation scaling, and ensembling - that lead to improved regularization. These designs are motivated by experiments showing that standard fully-connected output layers tend to rely on individual input neurons, which in turn do not cover the variance of the data. We call these two related phenomena *neuron dependency* and *expressivity*, propose different ways to measure them, and optimize the presented output layers for them. In our experiments, we compare these layer types for image classification and semantic segmentation across architectures, datasets, and application settings. We report significantly and consistently improved performance of up to 10% points in accuracy over standard output layers while reducing the number of trainable parameters by up to 90%. It is demonstrated that neither training of output layers is required, nor are output layers themselves crucial components of deep networks.

## 1 Introduction

Neural networks are powerful feature extractors that have become the standard approach for a myriad of tasks. New architectures are continuously introduced and set records on benchmark datasets (e.g. [24, 15, 44]). These networks differ in layer composition, depth/width or use specific concepts such as residual connections [15] or self-attention [34]. With growing capacity, their performance on large datasets tends to increase [44]. However, model complexity is also associated with overfitting, especially for small datasets where fine details of the training data are easily memorized [52, 2, 53].

Rather than defining another, possibly more complex architecture, we analyze what often remains unconsidered: the output layer. In image classification, networks usually end with a fully-connected (fc) layer that combines extracted features for the final output [43, 15, 17, 44]. As we will show, this layer is prone to overfitting since high dependencies on individual, possibly memorized features can arise. The same neurons are subsequently not able to generalize across examples. We call these two related phenomena *neuron dependency* and *expressivity* and illustrate a simplified example in Fig. 2.

Both problems can be improved by simple but effective changes to the output layer that require only few lines of code and achieve better generalization (i.e., better results on the test set [25], see e.g. Fig. 1). Those changes rely on four principles: activation scaling, fixed randomization, sparsity and in-layer ensembling (see Fig. 5). This work analyzes all layers in terms of their capability to reduce dependencies and/or increase expressivity. Then, the connection to network performance is shown through a comprehensive empirical study across datasets, architectures and application settings in image classification and segmentation. Furthermore, we investigate how stronger regularization

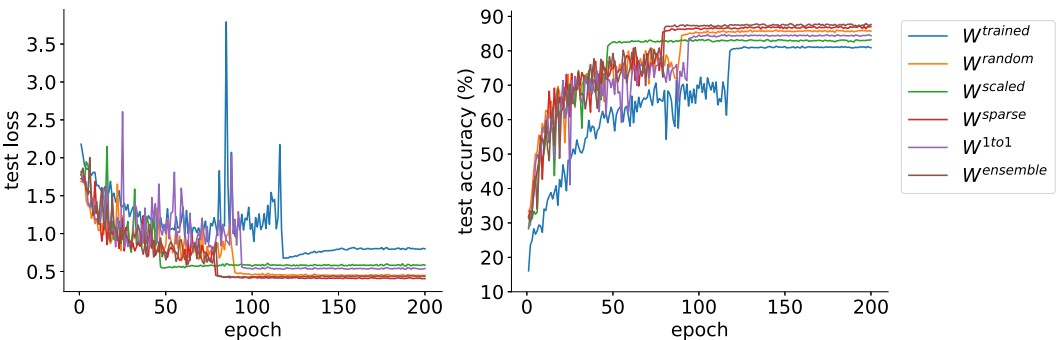

Figure 1: Effect of different output layer designs on cross-entropy loss (**left**) and accuracy (**right**) in a ResNet-50 for the STL-10 dataset. Best viewed in color.

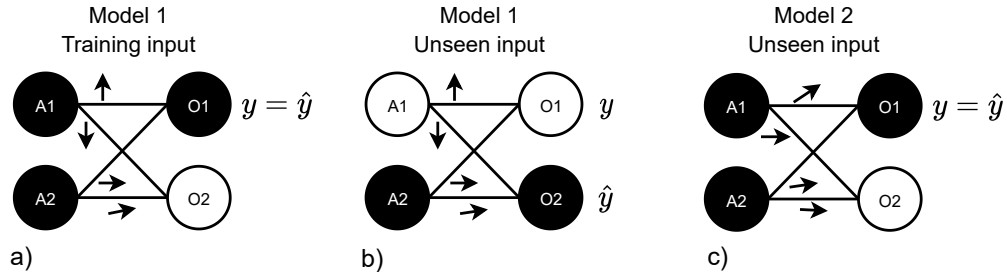

Figure 2: Schematic of neuron dependency/expressivity in fc output layers. The left side of each subfigure represents penultimate layer activations (A1-2), the right shows output neurons for each class (O1-2). Filled/blank circles indicate high/low activation, up-/downward facing arrows signal positive/negative weights. Higher activations of O1 lead to correct predictions in this example. Model 1 depends on neuron A1 to be activated to give high prediction scores to O1. This is the case for a training instance in **a)**. If Model 1 is applied to an unseen input pattern of same class in **b)**, higher scores are erroneously given to O2 since A1 remains inactive and A2 slightly favors O2. Model 1 fails to generalize as it depends on A1, which is not expressive enough to cover the variance of the target class. Instead, Model 2 shown in **c)** exhibits neurons with low dependency and high expressivity, where A1 generalizes to unseen patterns, while the activation of A2 can be regarded as backup. Note that this example is simplified and educational. See Sect. 3.3 for measurements.

can be induced by applying the identified principles to other parts of a network while reducing the computational footprint. In contrast to common practice, we find neither training of output layers to be necessary, nor that output layers are crucial components of deep networks. In summary, our **contributions** are:

- Introducing neuron dependency and expressivity as two factors contributing to overfitting and proposing ways to measure these factors

- Showing improved regularization of 5 different output layer designs up to 10% in absolute accuracy compared to standard fc layers and other common regularizers

- Empirical results showing that the proposed layers have improved dependency and expressivity, computational efficiency, wide applicability to both small and large datasets, extensibility to other parts of the network, and robustness in the choice of hyperparameters

## 2 Related Work

Regularization in deep learning is approached in various ways. Widely used methods are, e.g., normalization [19, 3], weight decay [32], data and adversarial augmentation [40, 1], early stopping [7], boosting [38], multitask learning [6], dropout [42], dropconnect [50], and Gaussian noise layers [10]. To the best of our knowledge, this is the first work that evaluates regularization with respect to

different output layer designs. Similar to dropout/dropconnect, output layers can be categorized as affecting the architecture according to the regularization taxonomy described in [25]. Unlike other regularizers, our methods are either hyperparameter-free or robust to their choice and can be applied to any deep net, including pre-trained ones that are less affected by overfitting (see Sect. 5.4 and 5.7).

Related to fixed randomization are the output layers used in [16, 39, 14], which show comparable performance to trained layers. One can also preallocate output layer weights with a defined structure [31, 16]. Besides output layers, weight fixing is for example applied to the first layer in the Extreme Learning Machine [18], or to different weight dimensions in [36]. In contrast, we omit hand-crafted weights, show improved regularization and relate to neuron dependencies. Further, we show that fixing or scaling the last conv block next to the output layer has a strong regularizing effect.

Sparsity is common in deep learning, e.g. the ReLU activation [13] or a $L_1$ penalty term in the loss function [46]. Sparsity has also been applied to the channels of Convolutional Neural Networks (CNNs) [8, 29]. Others induce sparsity by pruning connections before training under the lottery ticket hypothesis [11, 30], with the goal of reducing the number of parameters while not sacrificing performance [27, 45, 51]. Different to them, we show that (extreme) sparsity is not merely useful to improve computational efficiency, but to improve performance when applied to the output layer.

The Network in Network (NIN) [28] and All-CNN [41] both use global average pooling (GAP) followed by softmax, which replaces the fc output layer with an identify transform to simplify the network. This is further analyzed in [33]. We show its connection to neuron dependency/expressivity and achieve comparable or better performance on various datasets. Further, we observe that previous works do not leverage the full capacity of the last layer in modern networks, which enables the construction of computationally efficient in-layer ensembles that further boost performance in small and large datasets.

# 3 Neuron dependency and expressivity

## 3.1 Setting and notation

We consider neural networks consisting of an encoder $f_{enc} : \mathbf{X} \to \boldsymbol{a}$ followed by an output layer $f_{out} : \boldsymbol{a} \to \hat{\boldsymbol{y}}$. In this paper, the encoder is a CNN, taking as input an image $\mathbf{X} \in \mathbb{R}^{C \times H \times W}$ with $C, W$ and $H$ being input channels, width and height, respectively; and transforming it to a feature vector $\boldsymbol{a} \in \mathbb{R}^{1 \times N}$. Commonly in CNNs, 2D representations resulting from the final conv layer are aggregated by GAP [28] where $N$ corresponds to the number of pooled conv channels. The output layer transforms the embedding to output $\hat{\boldsymbol{y}} \in \mathbb{R}^K$ holding the probabilities of $K$ classes. The output layer is parameterized by a weight matrix $\boldsymbol{W} \in \mathbb{R}^{N \times K}$, and is commonly initialized as $\boldsymbol{W}^{random} \sim \mathcal{U}(-\sqrt{1/N}, \sqrt{1/N})$ [26]. Both $\hat{\boldsymbol{y}}$ and target $\boldsymbol{y}$ are used to compute the cross-entropy loss $\ell = -\sum_i^K y_i \log(\hat{y}_i)$. We use the terms features/channels/nodes or neurons interchangeably meaning activations $\boldsymbol{a}$. When required, we refer to individual instances with a superscript, e.g. $(\mathbf{X}^{(i)}, \boldsymbol{y}^{(i)}) \in \mathcal{D}$, with $\mathcal{D}$ being a dataset. Corresponding subsets are denoted as $\mathcal{D}_{train}$ and $\mathcal{D}_{test}$.

## 3.2 Concepts

During training, CNNs learn a set of visual patterns that are combined for a classification decision. However, if patterns remain undetected, e.g. due to noise in the image or inherent but unseen variance in the data, their activation values can become small and thus reduce the output values for the target class. When a network is overfitting, it learns malignant image-specific patterns by heart [52]. Such a network may depend on the activation of individual nodes, which in turn fail to generalize to patterns that are salient to a class. We call these two related phenomena neuron dependency and expressivity.

**Neuron dependency:** How much does a model depend on a single neuron? In a network with high neuron dependencies, output scores and thus performance drop significantly when certain neurons remain inactive. In contrast, a network with low neuron dependencies distributes activations across many neurons, so that a single inactive node does not have much influence on the classification.
**Neuron expressivity:** How much class-specific variance does a neuron cover? Neurons with low expressivity focus on unimportant details that do not characterize the properties of a class. In contrast, a neuron with high expressivity generalizes by activating to various patterns pertinent to a given class.

An example of neuron dependency/expressivity for a simplified fc output layer is illustrated in Fig. 2.

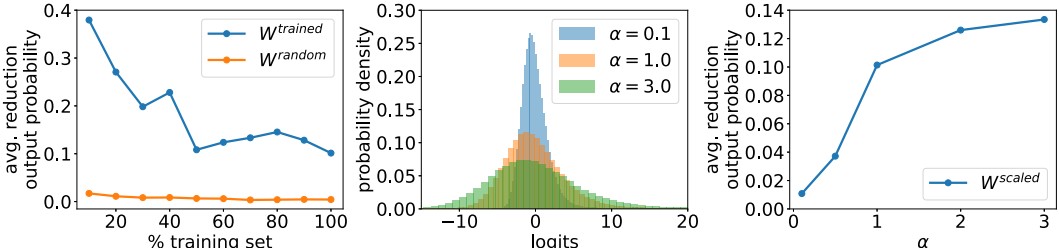

Figure 3: The effect of dataset size and activation scale on neuron dependency in a ResNet-50 trained on subsets of **CIFAR-100**, evaluated on the test set. **Left**: Small training sets lead to high neuron dependencies. **Center**: Scaling activations results in larger absolute logits. **Right**: Larger scales lead to higher dependencies. Best viewed in color.

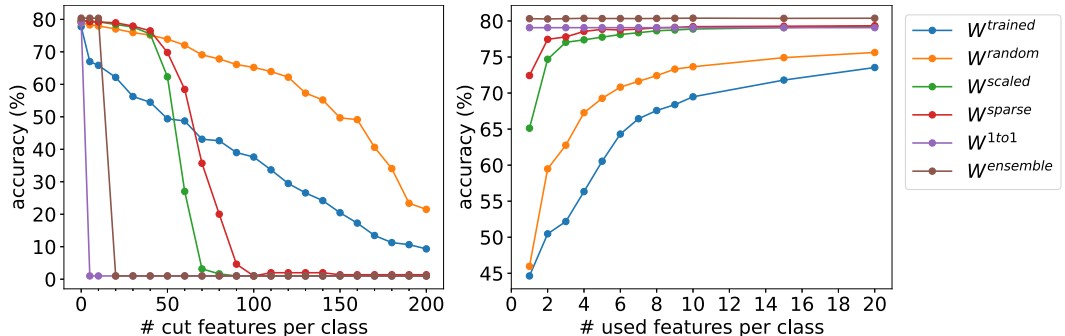

Figure 4: Neuron dependency (**left**) and expressivity (**right**) in a ResNet-50 with 2048 penultimate layer channels trained on **CIFAR-100** for different output layer designs, showing the change in accuracy on the test set. Best viewed in color.

### 3.3 Measuring dependency and expressivity

We introduce two ways of measuring dependency/expressivity: instance-based and class-based. The former is used to determine the dependency on the most important node for the predicted class given an instance. Importance scores for node $n$ and output class $\hat{k}$ are computed with Gradient$\odot$Activation [4], a global attribution method where we leverage the partial derivative of the softmax values:

$$a_n \frac{\partial \hat{y}_{\hat{k}}}{\partial a_n}. \tag{1}$$

Instance-based dependency is then measured as avg. reduction in output probabilities when ablating the most important feature w.r.t. the output class of any instance. This is illustrated for various training set sizes of CIFAR-100 [23] in Fig. 3 (left). With less data, fc output layers tend to depend more on single nodes. This is in contrast to class-based measures, which enable quantifying both dependency/expressivity and use various features jointly. Importances are determined for each class $k$ and over all test instances:

$$\sum_{i=1}^{|\mathcal{D}_{test}|} a_n^{(i)} \frac{\partial \hat{y}_k^{(i)}}{\partial a_n^{(i)}}. \tag{2}$$

Class-based dependency is then measured as drop in accuracy when ablating a given number of most important neurons per class. Measuring expressivity reverses this - the most important neurons per class are retained, all others are ablated. This is illustrated for both dependency/expressivity in Fig. 4. We see that standard (i.e. trained) fc output layers tend to depend on single channels to achieve high performance, but these very channels hold only limited class information.

## 4 Output Layer Types

We describe several simple output layer variants that require minimal changes to standard networks, decrease neuron dependency and/or increase neuron expressivity. All types are illustrated in Fig. 5.

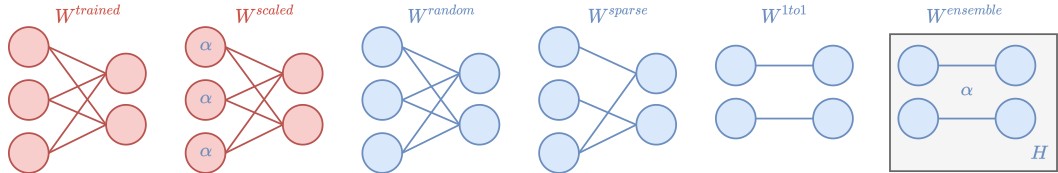

Figure 5: A visual comparison of various output layer types. Red/blue represent variable/fixed.

## 4.1 Standard output layers

The ubiquitous approach to compute class scores is to learn the parameters of a weight matrix $\boldsymbol{W}^{trained}$, s.t. $\hat{\boldsymbol{y}} = \sigma_{SM}(\boldsymbol{a}\boldsymbol{W}^{trained})$ with $\sigma_{SM}(\cdot)$ being softmax. Each feature is considered in the computation of each class score. As shown in Sect. 3, trained fc output layers can lead to high neuron dependencies, where the deletion of a single neuron might cause significant loss in performance, and low neuron expressivity, where multiple features are required for adequate predictions.

## 4.2 Scaled output layers

The reduction of an activation, e.g. due to changing light conditions, has a large influence on the output scores. This is simulated in Fig. 3 (center) by multiplying features during training with a scalar $\alpha > 0$, so that $\hat{\boldsymbol{y}} = \sigma_{SM}(\alpha\boldsymbol{a}\boldsymbol{W}^{scaled})$. Note that the variances of the output logit distributions increase with $\alpha$, resulting in larger differences (or smaller entropies) after softmax normalization. This results in greater dependencies of the model on individual neurons, as shown in Fig. 3 (right). However, if $\alpha$ is chosen small, the activations of individual neurons become insufficient for class discrimination with high confidence. The model is therefore forced to learn multiple class-specific features for each instance, which increases the expressivity of the neurons and also reduces their dependencies to some extent, as shown in Fig. 4. If not specified otherwise, we use $\alpha = 0.1$.

## 4.3 Random fixed layers

This setting uses $\boldsymbol{W}^{random}$ during training/inference, and its classification performance was first analyzed in [16]. The encoder learns to extract patterns that adjust to predetermined weights. Unlike activation scaling, the parameters are bounded and fixed to a small value range. For any class, the chosen uniform initialization is expected to assign similar weight values to multiple neurons, making them learn similar features. We suppose that the enforced similarity reduces dependency shown in Fig. 3 and 4 (both left), while small initialization values increase expressivity as in Sect. 4.2, shown in Fig. 4 (right).

## 4.4 Sparse fixed layers

In sparse output layers, class nodes use predetermined sets of channels, some of which might be shared across classes. First, a set of cutting indices $\mathcal{I}_k$ is randomly sampled for each class $k$, where sparsity is determined by the proportion $q$ of class-specific connections to cut, so that $|\mathcal{I}_k| = \lfloor qN \rfloor$ with $0 < q < 1$. Then, starting from a fixed random initialization as in Sect. 4.3, weights connecting to a given class are ablated so that: $\boldsymbol{W}_{i,k}^{sparse} = 0 \ \forall \ i \in \mathcal{I}_k$. Hyperparameter $q$ trades off dependency/expressivity. Larger values induce more sparsity, leading to greater dependencies to the remaining nodes, but forcing them to activate across instances, making them expressive. We set $q = 0.9$ in the experiments to show that high sparsity benefits generalization.

## 4.5 1-to-1 correspondence layers

The most extreme type of sparsity in an output layer is one with a single connection between a feature and a class. If these connections correspond to an identity transform, the activations of the penultimate layer are equivalent to the class logits - in practice, the output layer can hence be omitted. This was analyzed in [33, 28] and showed comparable results to a standard output layer. Formally, we have $\hat{\boldsymbol{y}} = \sigma_{SM}(\boldsymbol{a}\boldsymbol{W}^{1to1})$ with $\boldsymbol{a} \in \mathbb{R}^{1 \times K}$ and $\boldsymbol{W}^{1to1} \in \mathbb{R}^{K \times K}$, where $\boldsymbol{W}^{1to1} = diag(1, 1, \ldots, 1)$. In this layer, both the model's dependency on individual neurons as well as each neuron's expressivity are maximal. If a single neuron is ablated, the output logits for the class this neuron is connected to is

reduced to zero. However, individual neurons learn to cover the whole variance of a given class in the training set, which is one conjecture for their performance. Note that as mentioned in [33], we have the constraint $N = K$, which might be restrictive for small networks and large numbers of classes.

## 4.6 Ensemble layers

Is there a way to optimize for both low neuron dependency and high expressivity? Of the approaches discussed, 1-to-1 correspondence layers have the highest expressivity. Starting from this layer, a simple approach to reduce neuron dependency is to use the capacity of the penultimate layer and create multiple heads $h = 1 \ldots H$ with $N = KH$, each head being a 1-to-1 correspondence layer. Each head's output is effectively computed as $\hat{\boldsymbol{y}}^{\boldsymbol{h}} = \sigma_{SM}(\alpha \boldsymbol{a}^{\boldsymbol{h}})$ with $\boldsymbol{a}^{\boldsymbol{h}}$ being the activation part of head $h$. As in Sect 4.2, we introduce a scalar $\alpha$, which controls the magnitude of feature activations. For consistency, we denote this approach as $\boldsymbol{W}^{heads}$. The loss is computed for each head and averaged: $\frac{1}{H} \sum_{h=1}^{H} \ell(\hat{\boldsymbol{y}}^{\boldsymbol{h}}, \boldsymbol{y})$. Similarly, logits are averaged over heads for inference. Due to the induced redundancy, the performance only drops considerably after removing class-related neurons from all heads. In our experiments, we set $H$ to its maximum given any setting (architecture/dataset). Note that hyperparameter $\alpha$ in ensemble layers is the only one which is tuned to individual settings.

# 5 Experiments

We aim to show that the presented output layers from Sect. 4 outperform standard output layers and common regularization methods in various settings. Details about training, compute resources, code, datasets as well as additional experiments on dependency/expressivity are included in the appendix.

## 5.1 Small-scale and fine-grained classification

All layer types are first applied to small-scale and/or fine-grained classification, both of which are challenging and require regularization. Datasets include STL-10 (500 img/class) [9], CUB-200 ($\sim$30 img/class) [49], Cars-196 ($\sim$40 img/class) [22] and Food-101 (750 img/class) [5]. Table 1 shows results for the two popular backbones ResNet-50 [15] and DenseNet-169 [17], exchanging the output layer accordingly. In 53/56 settings, we see improved results over standard layers. Of these, 48 and 36 are significant with $p < 0.1$ and $p < 0.001$, respectively. Although there is no clear best method, it is worth noting that sparse and ensemble layers as enhancements of both random and 1-to-1 layers are significantly better ($p < 0.001$) in 7/8 settings, respectively. As expected, smaller performance differences are exhibited in Food-101, which is a considerably larger dataset, thus requiring less regularization. Among the worse settings, only 1 is significant ($p < 0.1$) for Food-101 since it involves strong regularization to multiple layers. These regularizers are discussed seperately in Sect. 5.5.

## 5.2 Large-scale classification and transfer learning

Machine learning models are subject to the bias-variance tradeoff [12], in which induced biases of the presented output layers might be too strong to fit the training data. We therefore want to shed light on how these layers behave in large-scale and transfer learning settings, where overfitting is less problematic. Datasets include CIFAR-100 (C100, 5000 img/class) [23], ImageNet (IN, $\sim$1200 img/class) from ILSVRC2012 [37] reported on the validation set, as well as CUB/Cars/Food with models being pre-trained on IN. Table 2 shows the results for the ResNet-50 backbone. In C100, we see consistent improvements with at least $p < 0.1$. On the other datasets, results are mostly comparable corroborating widespread applicability. It is worth mentioning that $W^{1to1}$ and $W^{ensemble}$ perform consistently better, and $W^{ensemble}$ significantly ($p < 0.1$) in multiple cases. With growing dataset sizes, both layers expose a strong constraint on the class neurons to fit an increasing number of examples. We believe this to be responsible for progressively separating the signal from the noise, leading to better generalization. On the other hand, neuron dependency is reduced in larger datasets (see Fig. 3 left) diminishing the effect of $W^{scale}$ and $W^{random}$. Moreover, $W^{random}$ and $W^{sparse}$ can be affected by predetermined feature-class weights that do not have to match features learned during pre-training, which might require larger adjustments to the weights of the last conv layer.

Table 1: Classification accuracy for different output layer designs in small-scale and fine-grained classification without pre-training. Exponent repeats describe probability values (*: $p < 0.1$, **: $p < 0.01$, ***: $p < 0.001$) indicating statistical significance based on a one-tailed normal approximation interval test comparing accuracy of the proposed layer designs to a baseline fc layer ($W^{trained}$). Symbols $*$ and $\dagger$ denote better/worse performance than baseline, respectively. **Bold** denotes best performance.

| | | STL-10 | CUB-200 | Cars-196 | Food-101 |
|---|---|---|---|---|---|
| ResNet-50 | $W^{trained}$ (baseline) | 81.36 | 57.18 | 81.20 | 83.70 |
| | $W^{scaled}$ | 83.33* | 63.46*** | 87.07*** | **85.46*** |
| | $W^{scaled}$ block | 86.42*** | 66.74*** | **87.53*** | 85.03** |
| | $W^{random}$ | 86.08*** | 60.91** | 83.02* | 84.20 |
| | $W^{random}$ block | 86.59*** | **67.21*** | 84.07*** | 84.04 |
| | $W^{sparse}$ | 87.23*** | 66.27*** | 85.47*** | 85.45*** |
| | $W^{1to1}$ | 84.78*** | 58.56 | 80.51 | 84.41* |
| | $W^{ensemble}$ | **87.94*** | 62.98*** | 85.76*** | 85.36*** |
| DenseNet-169 | $W^{trained}$ (baseline) | 81.88 | 55.33 | 80.82 | 84.31 |
| | $W^{scaled}$ | 86.53*** | 63.31*** | **85.85*** | 85.05* |
| | $W^{scaled}$ block | 85.89*** | 65.57*** | 85.35*** | **85.44** |
| | $W^{random}$ | 86.11*** | 61.24*** | 83.52** | 84.90* |
| | $W^{random}$ block | 86.64*** | **65.99*** | 82.93* | 83.25† |
| | $W^{sparse}$ | 86.58*** | 62.75*** | 85.79*** | 84.63 |
| | $W^{1to1}$ | 86.06*** | 55.37 | 83.75** | 84.11 |
| | $W^{ensemble}$ | **87.00*** | 64.15*** | 85.09*** | 84.91* |

Table 2: Classification results for different output layer designs in large-scale image recognition and transfer learning. + denotes fine-tuning from ImageNet. See Table 1 for other symbols.

| | C100 | IN-top1 | IN-top5 | CUB-200+ | Cars-196+ | Food-101+ |
|---|---|---|---|---|---|---|
| $W^{trained}$ | 77.75 | 76.36 | 93.12 | 80.91 | 91.73 | 87.32 |
| $W^{scaled}$ | 79.65* | 76.08 | 92.84 | 78.68† | 90.91† | 87.21 |
| $W^{random}$ | 78.91* | 76.08 | 93.15 | 80.89 | 91.72 | 87.29 |
| $W^{sparse}$ | 79.46* | 75.32†† | 92.36††† | 80.38 | 92.07 | 87.31 |
| $W^{1to1}$ | 79.07* | 76.53 | 93.32 | 81.79 | 91.87 | 87.31 |
| $W^{ensemble}$ | **80.38*** | **76.62** | **93.46*** | **82.22*** | **92.77*** | **87.76** |

## 5.3 Use Case: Medical imaging

Output layer design is critical in fields such as medical imaging, which presents special challenges to regularization: Datasets tend to be small, imbalanced, abnormalities might fill only a few pixels of the image, and appearances between classes are often similar. In addition, transfer learning with IN weights is either inaccessible due to architectural differences (e.g. image segmentation, 3D Magnetic Resonance Imaging) or less effective due to large domain differences. This is first illustrated on the APTOS Kaggle challenge dataset (3662 images, 193-1805 img/class) [20], with the goal of detecting diabetic retinopathy severities in retinal fundus images. We use the public training dataset to train a multi-class classifier and perform 5-fold cross-validation. Table 3 shows the results. We consistently get better performance with regularization and reduce the gap to a pre-trained network. Furthermore, an additional experiment in the appendix indicates that the standard output layer is biased towards the prevalent class, which is inherently remedied through randomization.

We provide further evidence that the proposed layer designs positively affect tasks other than classification. We learn a U-Net [35] for binary semantic slice-based segmentation of Computed Tomography scans of livers comparing a standard 1x1 conv output layer with 64 parameters to both a fixed randomized and an ensemble layer. Due to the limited number of parameters, we omit $W^{scale}$ and $W^{sparse}$ here. Different to classification, the output of a U-Net itself can be interpreted as a 1-to-1 layer. One can still build an ensemble by treating each output channel as a head. Both $W^{random}$ and $W^{ensemble}$ ($H = 10$) are then applied to the CHAOS [21] and SLIVER [47] datasets. For CHAOS,

Table 3: Quadratic weighted kappa and accuracy (with significance) for different output layers in ResNet-50 for the APTOS dataset. + denotes fine-tuning from IN. See Table 1 for other symbols.

| | Kappa | Acc. |
|---|---|---|
| $W^{trained}$ | 0.816 | 77.44 |
| $W^{scaled}$ | 0.818 | 78.38 |
| $W^{random}$ | 0.848 | 79.32* |
| $W^{sparse}$ | 0.840 | 79.60* |
| $W^{1to1}$ | 0.856 | 80.08* |
| $W^{ensemble}$ | **0.866** | **80.78**\*\* |
| $W^{trained}+$ | 0.909 | 85.02 |
| $W^{sparse}+$ | 0.910 | 85.17 |
| $W^{ensemble}+$ | **0.912** | **85.56** |

Table 4: Jaccard coefficients in segmentation

| | CHAOS | SLIVER |
|---|---|---|
| $W^{trained}$ | 0.77 | 0.83 |
| $W^{random}$ | **0.80** | 0.85 |
| $W^{ensemble}$ | 0.78 | **0.86** |

Table 5: Regularization comparison

| | STL | CUB | CUB+ | Cars |
|---|---|---|---|---|
| Dropout [42] | 82.73 | 63.20 | 80.26 | 83.98 |
| Dropconn. [50] | 86.15 | 61.48 | 80.41 | 85.06 |
| Add. Noise [10] | 82.51 | 52.74 | 80.91 | 76.77 |
| $W^{trained}$ | 81.36 | 57.18 | 80.91 | 81.20 |
| $W^{sparse}$ | 87.23 | **66.27** | 80.38 | 85.47 |
| $W^{ensemble}$ | **87.94** | 62.98 | **82.22** | **85.76** |

we train on 15 randomly selected patients (2155 slices) and evaluate on the remaining 5 (719 slices). We then test for generalization by training on all 20 patients from CHAOS and evaluating on the external SLIVER dataset consisting of 20 patients (4159 slices). Table 4 shows improved results in both settings.

## 5.4 Other regularization techniques

Table 5 compares our most competitive methods to other popular regularizers when applied to a standard output layer. Nodes/connections in dropout/dropconnect are both removed with $p = 0.7$, and the noise layer adds a Gaussian with $\mu = 0$ and $\sigma = 0.1$ before applying the fc layer. In all cases, our variants perform better. Whereas noise does not benefit training here, dropout/-connect is supporting regularization. However, both of the latter methods come with two main disadvantages. First, they add complexity by changing states in each iteration and having different behavior during training and inference. Second, hyperparameter tuning is necessary, while our layers are either hyperparameter-free or stable to them. See the ablation study in Sect. 5.7 for evidence.

## 5.5 Beyond output layers - block scaling and randomization

Activation scaling and randomization are techniques applicable to any layer and increase regularization further. This is demonstrated for ResNet-50 and DenseNet-169 in Table 1. Both architectures consist of multiple blocks, each holding groups of conv layer, batch normalization (BN) and activation function. For $W^{random}$ block, all layers of the last block and the output layer are kept in their initialized state during training. Similarly, in $W^{scaled}$ block, activations of all layer groups in the last block are scaled during training. In ResNet-50, block scaling outperforms output layer scaling in 3/4 datasets by up to 3% points, and block randomization increases performance in 3/4 datasets by up to 6% points compared to output layer randomization. In DenseNet-169, block scaling outperforms output layer scaling in 2/4 datasets by up to 2% points, and block randomization increases performance in 2/4 datasets by up to 4% points compared to output layer randomization. Only in Food-101 and DenseNet, block randomization performs significantly worse than baseline because regularization is too strong leading to underfitting ($train\ loss = 0.47$ compared to $0.02$ in $W^{random}$).

## 5.6 Computational efficiency

The design of the head of deep CNNs has a great impact on computational efficiency. Standard output layers alone can contain a large amount of parameters, as CNNs typically hold more channels as they get deeper and the number of classes can become large. In ImageNet and a ResNet-50, for example, the output layer alone generates over 2 million parameters, which are saved in $W^{1to1}$ and $W^{ensemble}$. This problem compounds when using multiple fc layers. In a VGG-16, for instance, 3 fc layers are employed after the last conv layer. As Table 6 shows, omitting all fc layers saves up to 90% in parameters, a considerable amount of memory, and time for a forward/backward pass while

Table 6: Computational efficiency comparison in CUB-200 highlighting that the number of trainable parameters can often be reduced while accuracy is improved. + denotes fine-tuning from ImageNet.

| Architecture | #Params in M. | Mem.[GB] | GFLOPS | Time [ms/it.] | Accuracy |
|---|---|---|---|---|---|
| VGG16 $W^{trained+}$ | 135.1 | 7.5 | 31.1 | 114 | 78.68 |
| VGG16 $W^{1to1+}$ | 13.5 | 5.9 | 30.4 | 106 | 79.27 |
| VGG16 $W^{ensemble+}$ | 14.7 | 5.9 | 30.8 | 106 | 81.15** |
| Res50 $W^{trained}$ | 23.9 | 5.1 | 8.2 | 71 | 57.18 |
| Res50 $W^{random}$ block | 8.5 | 5.0 | 8.2 | 67 | 67.21*** |

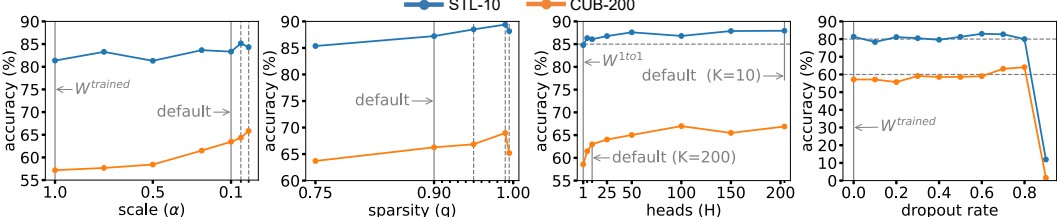

Figure 6: Ablation study showing stability and consistency of our output layer designs

increasing accuracy. If a ResNet-50 is used, randomization of the last conv block next to the output layer yields savings of about 65% in trainable parameters while increasing accuracy by 10% points.

## 5.7 Ablation study

Note that $W^{random}$ and $W^{1to1}$ are hyperparameter-free compared to other regularizers such as dropout/-connect, thus saving the cost of tuning them. Although other layer variants possess hyperparameters, we show in Fig. 6 for the ResNet backbone that they are stable (no large jumps in vicinity) and consistent (tend to monotonicity w.r.t. performance). In $W^{sparse}$, the maximum accuracy for both datasets is at $q = 0.99$ (20 nodes per class) and drops only slightly for $q = 0.995$. Similarly, downscaling in $W^{scale}$ improves performance at a small cost if the optimum is not hit. Also, more heads in $W^{ensemble}$ tend to increase performance. What is the result of adding more heads than given by the constraint $N = KH$? If $N < KH$, which is the case for CUB-200 and $H > 10$, we add an additional 1x1 conv layer, BN and ReLU with $KH$ nodes to adjust for the missing channels. Although this leads to a considerable increase in parameters ($NKH$ for the conv layer), it helps with generalization, contradicting the common belief that overparameterization leads to overfitting [48]. In contrast, dropout is not stable or consistent. With a dropout rate of $0.9$, the network fails to train in both datasets. Furthermore, the optimum for CUB lies at $0.8$, the same hyperparameter choice in STL would result in worse performance than baseline.

## 6 Conclusion

In this work, we introduced neuron dependency and expressivity as factors contributing to overfitting. Then, different output layers were defined to optimize both and showed improved regularization in various settings while being efficient and robust to hyperparameters. Although these layers are simple, they have high practical relevance due to the importance of regularization and the ubiquity of output layers in deep nets. In addition to their application, they may also be useful as primitives in future (automatically created) architectures. Although improving regularization, we note that optimizing for neuron dependencies/expressivity does not *solve* overfitting. For example, an unknown or noisy instance may result in reduced activations in the majority of nodes in the penultimate layer. Finally, we speculate that overfitting may not just be a function of the number of parameters in the encoder. Instead, it might be more important how the extracted features are combined in the output layer.

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
