# OpenReview forum: "The Regularizing Effect of Different Output Layer Designs in Deep Neural Networks"
_NeurIPS.cc/2021/Conference — NeurIPS 2021 Submitted_

### Official Review · Reviewer_V64q · 2021-07-15

**Rating:** 6
**Confidence:** 4

**Summary:**

This experimental work studies various sparse output-layer types for transfer-learning. Authors report interesting gains through usage of certain learned/fixed output layers. Although I found the experiments interesting and believe that they would spark more research on the topic; the story and results needs a little bit more work.

**Ethics Review Area:**

["I don’t know"]

**Limitations And Societal Impact:**

Authors talk about the limitations at the end of the paper, which I appreciate.

**Main Review:**

# Strengths
- The correlation between neuron saliency and results are very interesting and novel as far as I am aware.
- Usage of sparse output layers for transfer also novel as far as I know.

# Limitations/Questions/Recommendations
0. It seems like authors use weight decay value of 0.0001 (seems to be same for backbone and output layer). This choice affects the performance of dense output layers greatly. Therefore needs to be searched (possibly only for the output layer). It would be great to have a curve showing the performance of dense baseline output layer with different regularisation coefficients on x-axis.
1. I think the "scaling head" corresponds to using a temperature parameter in softmax (https://en.wikipedia.org/wiki/Softmax_function). Assuming that it is case, I think this connection must be clear and this variant should be studied under "loss type" used, as this is one parameter one can pick during finetuning (for all different architectures).
2. Figure 2 didn't really helped me to understand the terms introduced neuron dependency/expressivity better. The formulas presented later did though. I recommend authors to use the definition earlier. More importantly I think authors should use existing terms instead of introducing new ones. "neuron dependency" means "saliency" (as cited in text). Similarly I think neuron expressivity measure can be captured with 'per-class saliency' or "class-saliency". I think the introduction can be shorten significantly with these changes helping reading to focus on main results.
3. Can the ensemble connections be represented by a single sparse layer? It would be nice to discuss connections between random sparse layers, 1-to-1 layers and ensemble layers. To me the later 2 seems to be a specific case of a random sparse layer. Furthermore it is not clear why authors choose to report values for 90% for the sparse layers, despite the fact higher sparsities seem to give better results.

## Minor Points
- It would be better if References to accuracy gain (i.e. 10%) in abstract and intro has more detail in them like up and downstream datasets.
- Having 3 input neurons in Figure-5 for 1-to-1 and ensemble layers can make them easier to understand.
- Personally I find the p-scores confusing compared to confidence intervals or variance; which are used more commonly in ML (from what I read/see). Is there a reason this is preferred. If so, I think it would be nice to have a sentence on this in the main text.

## After Rebuttal
I like to thank authors for their response, especially for spending the time to create the weight decay ablation. It is helpful to see and I recommend authors to use the values found in the next version of the paper, even if the conclusions are not changing. Another recommendation is to have an l-1 penalized dense output layer baseline.

I decided to keep my score as it is, which is a "weak-accept". I think the paper would become a clear accept with further experiments (i.e. more backbones/down-stream tasks like transformers and NLP) and ablation studies (like different levels of sparsity).


**Time Spent Reviewing:**

4.5

---

> ### Author Response · Authors · 2021-08-09
> **Thanks and response to concerns**
>
> Thank you for your detailed review. We address your questions and concerns in the following.
>
> >It would be great to have a curve showing the performance of dense baseline output layer with different regularisation coefficients on x-axis.
>
> Weight decay of 0.0001 is used across settings and layers as in [1,2].
> We plot the performance of different weight decay values in a dense baseline output layer for two exemplary datasets and ResNet-50.
>
> STL-10: https://imgur.com/a/WJHhrI7
>
> CIFAR-100: https://imgur.com/a/h4gZKkc
>
> None of these choices perform better than our weakest output layer designs. Initially, we also tested weight scaling and weight clamping, but activation scaling performed better among these options.
> We added this experiment and the discussion of results to the appendix.
>
> >I think the "scaling head" corresponds to using a temperature parameter in softmax
>
> Thanks for pointing this out. Alpha corresponds to a softmax temperature in $W^{ensemble}$. We added this information to Sect. 4.6.
>
> >...should be studied under "loss type" used, as this is one parameter one can pick during finetuning
>
> We considered 0.5, 1., and 2. as choices for alpha in $W^{ensemble}$ while tuning on a small portion of the training data. We found that in non-pretrained settings (including ImageNet), 1.0 and 0.5 work best and perform similarly well (as shortly mentioned in Sect. A.1). Because of the similarity with $W^{1to1}$, where we implicitly set alpha to 1, the performances presented in Table 1 are comparable. In particular, $W^{ensemble}$ works better than $W^{1to1}$ in all these cases, i.e., the statement that using an ensemble improves the results should still be valid.
>
> >Figure 2 didn't really helped me to understand the terms introduced neuron dependency/expressivity better. The formulas presented later did though.
>
> We are glad that one of the explanations helped. We included several viewpoints (high-level definition + minimum viable example + formulae) to ensure that at least one would lead to understanding of these concepts among a diverse group of readers.
>
> >I think the introduction can be shorten significantly with these changes helping reading to focus on main results.
>
> We feel that 30 lines (17-47) is reasonable and further shortening the introduction might make it difficult to motivate the idea behind the paper.
>
> >I think authors should use existing terms
>
> We added the term and relation to saliency to make it clearer for the community.
> However, we believe that using this term exclusively could lead to misunderstandings, since we use dependency both for individual instances as well as at the class level. Moreover, both what we call class-based dependency and class-based expressivity would thus map to ‘per-class saliency’, which would be problematic since they measure different (albeit related) things.
>
> >Can the ensemble connections be represented by a single sparse layer? It would be nice to discuss connections between random sparse layers, 1-to-1 layers and ensemble layers.
>
> All three are related to each other. $W^{1to1}$ is a special case of $W^{sparse}$, where all weights are set to 1 and sparsity is both directed and maximized in the sense that there is a single connection between a penultimate layer node and a class node. In other words, an activation of the penultimate layer corresponds to the activation of a class.
> $W^{ensemble}$ uses multiple heads. Each head represents a 1-to-1 layer and thus a classifier in the ensemble. Each head provides a prediction for which an individual loss is computed, and all losses are averaged during training.
> Thus, in a strict sense, $W^{ensemble}$ cannot be directly called a sparse random layer, since multiple losses are used and the activations of each 1-to-1 head are independent of each other. However, $W^{sparse}$ could also be defined to split its output into multiple heads. In this case, $W^{ensemble}$ would again be a special case.
> We adjusted Sect. 4.5 and 4.6 to make the differences/similarities clearer.
>
> >Furthermore it is not clear why authors choose to report values for 90% for the sparse layers, despite the fact higher sparsities seem to give better results.
>
> Initially, we didn’t do a fine-grained search (i.e. from 0.9 to 0.995 as done in Fig. 6) for the best value of $q$ over all datasets. Most important for us was to show that
> * sparsity has an impact on neuron dependency/expressivity (saliency)
> * that we don’t have to do a long and individual search for any architecture/dataset
> * that it benefits regularization
>
> The chosen value of $q=0.9$ did the job. However, in Section 5.7 we were interested in how far we can go, and the results suggest that values above 0.9 work even better (in particular $q=0.99$).
>
> >It would be better if References to accuracy gain (i.e. 10%) in abstract and intro has more detail in them like up and downstream datasets.
>
> We added that this number corresponds to a non-pretrained setting and that the proposed layers perform mostly on par in downstream (pretrained) settings compared to baseline.
>
> >Having 3 input neurons in Figure-5 for 1-to-1 and ensemble layers can make them easier to understand.
>
> We agree. 1 additional input neuron has been added to both layers in Fig. 5.
>
> >Personally I find the p-scores confusing compared to confidence intervals or variance; which are used more commonly in ML (from what I read/see). Is there a reason this is preferred?
>
> This was more pragmatic due to computational complexity, since we had to perform experiments for each architecture, for all proposed layers, and for each dataset (including ImageNet, which takes a lot of time to train with our hardware).
>
> Thank you again for your feedback and help in improving this paper.
>
> ---
> References:
>
> [1] He, Kaiming, et al. "Deep residual learning for image recognition." Proceedings of the IEEE conference on computer vision and pattern recognition. 2016.
>
> [2] Huang, Gao, et al. "Densely connected convolutional networks." Proceedings of the IEEE conference on computer vision and pattern recognition. 2017.

---

### Official Review · Reviewer_MAam · 2021-07-16

**Rating:** 3
**Confidence:** 4

**Summary:**

Usually a fully connected layer is used as the final layer of an image classification network. To reduce the overfitting of this layer, this paper studies the use of activation scaling, fixed randomization, sparsity and in-layer ensembling as a regularization technique.
This paper also introduces neuron dependency and expressivity as two factors contributing to overfitting and proposes ways to measure these factors.


**Ethical Concerns:**

I don't have nay concern related to ethical issues.



**Limitations And Societal Impact:**

There is no discussion on potential negative societal impact.

**Main Review:**

- There are no evaluation results on the popular classification benchmarks such as ImageNet, CIFAR10 or CIFAR-100. Without results on these benchmarks it’s hard to know if the proposed method gives additional benefit in comparison to the current state-of-the-art regularization methods. Specially, it is important that the paper includes results on the large scale dataset of ImageNet as the results may change as we have more training data.
- The current-state-of-the-art results on the provided benchmarks is much higher than the reported results in the paper. It would be helpful if authors provide results on top of stronger networks.
- By adding more regularization to the previous layers of the network, we may need less regularization for the output layer. I think authors should also provide analysis for their proposed method when they apply it on top of other common regularization and also data-augmentation methods.
- The writing of the paper is not clear and needs improvement.
- Table 5 provides a comparison to the dropout and shows that ensemble method which builds multiple heads on top of the network has a better performance in comparison to the dropout. Are the proposed ensemble method and dropout orthogonal and can we combine them together?


**Time Spent Reviewing:**

1

---

> ### Author Response · Authors · 2021-08-09
> **Thanks and response to concerns**
>
> Thank you for your detailed feedback. We provide answers to your main concerns in the following.
>
> >There are no evaluation results on the popular classification benchmarks such as ImageNet, CIFAR10 or CIFAR-100.
>
> Section 5.2 and Table 2 present experiments and discussions on large-scale classification and transfer learning. In particular, the results for ImageNet and CIFAR-100 are given. In addition, the neuron dependency/expressivity results for CIFAR-100 are presented in Figs. 3 and 4. In total, experiments with 9 different datasets and 4 different architectures are presented.
>
> >It would be helpful if authors provide results on top of stronger networks.
>
> We included datasets that are prone to overfitting, and mainly selected architectures that we thought were strong enough to overfit in these settings to have a good way to compare between layers.
>
> It is likely that stronger models would lead to even worse overfitting in the baseline. For example, in Table 9 of the appendix, we see that a ResNet with 3 blocks instead of 4 (i.e. a simpler net) leads to better performance in small datasets.
>
> >By adding more regularization to the previous layers of the network, we may need less regularization for the output layer. I think authors should also provide analysis for their proposed method when they apply it on top of other common regularization and also data-augmentation methods.
>
> We initially did not combine with other regularization methods such as dropout/-connect because we intended to compare the proposed output layers with existing regularizers. However, we use weight decay and common data augmentation (A.1 appendix).
>
> In the following, we want to shed light on the effect of adding dropout regularization to previous layers and extended Sect. 5.4 with results of the following experiment.
> Specifically, we add dropout after each conv block of ResNet-50 (i.e. 4x dropout). A grid search is applied for the dropout rate (0.1, 0.2, 0.5, 0.7) on a small subset of the training data.
> For STL-10, a dropout rate of 0.1 works best and achieves an accuracy of 85.63 (compared to 81.36 in baseline). In comparison, $W^{sparse}$ and $W^{ensemble}$ regularizers achieve better accuracies of 87.23 and 87.94, respectively. Applying $W^{sparse}$/$W^{ensemble}$ on top of this regularization scheme does not improve the results (85.41/86.73, respectively).
> In CUB-200 without pretraining, a dropout rate of 0.2 works best but achieves worse performance (55.22 vs. 57.18 in baseline).
>
> These results suggest that the proposed layers are good and stable standalone regularizers. Also note that we can get even stronger regularization without other regularizers by fixing or scaling layers of the encoder as shown in Sect. 5.5.
>
> >Are the proposed ensemble method and dropout orthogonal and can we combine them together?
>
> Ensemble layers are an enhancement of 1-to-1 layers. In the latter, dropout would not be beneficial as the node representing the correct class may be set to 0 at random, leading to high losses.
> However, ensemble layers use multiple 1-to-1 layers and may still be able to fit the training data after ablating individual class nodes through dropout. We therefore conducted another experiment on STL-10/CUB-200 and ResNet-50 to test this idea.
> In this case, large dropout values of 0.7 work best (which is the same best value when applying dropout to the baseline output layer). In CUB200, $W^{ensemble}$+dropout achieves a better accuracy of 64.92 compared to 62.98 in $W^{ensemble}$. In STL-10, results are comparable with 87.77 in $W^{ensemble}$+dropout vs. 87.94 in $W^{ensemble}$ only.
>
> This suggests that adding dropout to $W^{ensemble}$ might help in some cases. However, this goes along with tuning another hyperparameter and the theoretical possibility that correct class nodes of all heads are ablated as in 1-to-1 layers.
>
> We added the results of this experiment to Sect. 5.4.
>
> >There is no discussion on potential negative societal impact.
>
> We argue in 1c) of the checklist about this.
>
> >The writing of the paper is not clear and needs improvement.
>
> We tried our best to make this paper easy to follow, but we welcome suggestions on improving it.
>
> Thank you again for your feedback and help in improving this paper.

---

### Official Review · Reviewer_kZyP · 2021-07-19

**Rating:** 6
**Confidence:** 5

**Summary:**

This paper investigates the regularizing effect of different output layer designs in deep neural networks. The authors propose several different design choices of the output layers, which are all very simple but effective as the authors demonstrated in the experiments. They also propose two terms named neural dependency and expressivity between the neurons and classes. With these simple designs of the different output layers, the experiments on multiple classification tasks and resnet/densenet CNN models improve the performance for a large margin, which helps avoid the overfitting problem.

The main contributions:
1. Propose simple yet effective output layer designs for deep neural networks.
2. Introduce neural dependency and expressivity factors to measure the overfitting problem.
3. Experiments on different classification tasks are expressive.

**Main Review:**

This paper studies the overfitting problem and the regularization methods for deep neural networks. Different from previous work, the authors work on the output layer and specifically design different output layers for regularization. They introduce five different simple methods of the output layers. Easy to follow and interesting analysis, the introduced two factors are also some evidence for their analysis. The experiments are also express to see the improvements. The paper is also very clear to follow.
However, there are also some questions and concerns:
1. The alpha in the scaled output layers is designed to smooth the output distributions as I understand, though the authors connect to the dependencies. Therefore, it is expected alpha has huge effects on the model training, can you plot the curves of the performance change? Btw, the choice of alpha also depends on the different datasets since the data (class) distribution of each task should be different. It is expected to see the comparisons.
2. Similarly, the choice of q is also interesting, could the authors also study the effect of q? Too many zero connections hurt the performance badly. For sparse fixed layers, could you give a clear comparison between this to the dropout and dropconnect method? Strong relationships, is sparse version a static version of dropout?
3. Could you give some clear explanations or visualizations about the 1to1 layer? What's the most important impact of this design? At least, this does not change the output distributions.
4. One major weakness is that the method is only compared with the basic resnet and densenet models, and also the basic training, without comparison to advanced methods/models. This significantly reduces the importance since this paper tries to sell the output design, which should be applied together with other stronger models and regularization methods to see the importance and effectiveness.
5. Final minor one, In table 2, IN-top1 is only 76.36% for baseline. However, it seems that the Resnet-50 can achieve about 77.15% as shown in https://paperswithcode.com/sota/image-classification-on-imagenet. Is there some difference?

Overall, I like this paper and am pleased to improve the score with answers.

**Time Spent Reviewing:**

4 hours

---

> ### Author Response · Authors · 2021-08-09
> **Thanks and response to concerns**
>
> We are happy that you like the paper and want to thank you for your detailed review. We provide answers to your main concerns in the following.
>
> > The alpha in the scaled output layers is designed to smooth the output distributions as I understand, though the authors connect to the dependencies.
>
> In Sect. 4.2. and Fig. 2, we take the viewpoint of an overfitting network, which we try to clarify with another example. Consider there is a most important feature for an instance - e.g. a memorized feature extracted from the background that leads to a high activation value + a high weight connection to the output class + low weight connections to other classes - which is ablated during testing, e.g. because this memorized feature no longer occurs.
>
> In this case, a baseline output layer (with $\alpha=1$) will cause a larger drop in respective class probability than a model where the penultimate layer activations are downscaled, because individual importances are suppressed. This means that $W^{scaled}$ with small alpha is less dependent on individual activations.
>
> However, the viewpoint of smoothing the output distribution is valid, too. If penultimate layer activations are downscaled and the weights of the output layer remain unchanged compared to baseline, then the output distribution is smoothed. Theoretically, the weights of the output layer could adapt during training. If they would adapt inversely (e.g. weights end up being multiplied by 10 for $\alpha=0.1$ compared to a baseline model), the output distribution would be the same. In practice, we see that this is actually not happening and the logits scale with $\alpha$ (Fig. 3 center). In other words, the output distribution gets smoothed in practice.
>
> We incorporated this perspective more clearly in Sect. 4.2 to improve comprehensibility.
>
> >Therefore, it is expected alpha has huge effects on the model training, can you plot the curves of the performance change?
>
> $\alpha$ has a large effect on performance. For instance, Fig. 1 (right) shows that accuracy is increasing faster in $W^{scaled}$ than in $W^{trained}$ and ends up in an overall larger accuracy at the end of training. Table 1 provides further evidence that $W^{scaled}$ has a large effect in various datasets prone to overfitting. In Sect. 5.7 and Fig. 6 in particular, the effect of $\alpha$ is further characterized where we plot performances at different $\alpha$ values. It shows that smaller values increase accuracy by a large margin.
>
> >Btw, the choice of alpha also depends on the different datasets since the data (class) distribution of each task should be different.
>
> This is correct. Given a static setting, where activations and weights are fixed, the choice of $\alpha$ would impact the distributions for a different number of classes. In general, the number of classes affects the output distribution in standard fc layers, even if the sum of logits remains the same.
> In practice, we show in Fig. 6 for STL-10 and CUB-200 that smaller values for $\alpha$ tend to increase performance irrespective of the number of classes (10 vs. 200).
> We chose $\alpha=0.1$ for all other experiments to show that hyperparameter tuning is not required in practice to improve the baseline.
> We added this detail to Sect. 4.2.
>
> >Similarly, the choice of q is also interesting, could the authors also study the effect of q? Too many zero connections hurt the performance badly.
>
> Yes, we studied the effect of the choice of $q$ in Sect. 5.7 and Fig. 6. We figured that many zero connections help the model to perform better. In fact, cutting 99% of all connections at random achieved best performance in both STL-10 and CUB-200. Only when zeroing 99.5%, which corresponds to 10 connections per class in ResNet-50, performance drops slightly.
>
> >For sparse fixed layers, could you give a clear comparison between this to the dropout and dropconnect method? Is sparse version a static version of dropout?
>
> Dropout sets activations to 0 at random in each iteration and uses all activations during inference but rescales the weights. Dropconnect instead sets connections to 0 at random in each iteration. Dropout and dropconnect change states in each iteration, have different training/test behavior, and weights are not fixed.
>
> $W^{sparse}$, on the other hand, uses all activations, fixes all connections to their initialized values and sets some random connections initially to 0. Also training/test behavior is the same. One could argue that $W^{sparse}$ is, to some extent, a static version of dropconnect both in terms of not adapting weights and only choosing initially which weights to drop.
>
> Empirically, $W^{sparse}$ obtains better classification results than dropout and dropconnect as shown in Table 5. In addition, $W^{sparse}$ is not prone to hyperparameter choices compared to dropout, which is shown in Sect. 5.7.
>
> We added a discussion about these differences to Sect. 4.4.
>
> >Could you give some clear explanations about the 1to1 layer? What's the most important impact of this design?
>
> A 1-to-1 layer is not a layer in the conventional sense, since it is an identity transform of the last conv layer’s activations. As noted in Sect. 4.5, it is an extreme type of sparsity. In fact, it is a special case of $W^{sparse}$ where connections are not dropped at random and there is only one connection per class without overlaps.
> This design has several implications:
> * Both neuron dependency and expressivity are maximized. Regarding the former, ablating a class connection turns the corresponding class logit to zero. For the latter, a single neuron has to activate strongly and generalize to all class instances of the training set.
> * It (also) works well in large-scale and fine-tuning settings (Table 2). As reasoned in Sect. 5.2., it exposes a strong constraint on the class neurons to fit a large number of examples, which might be responsible to separate signal from noise instead of memorizing the latter, therefore might better generalize to test instances as well
> * Benefit in terms of interpretability: it does not require CAM/Grad-CAM [1,2] to visualize patterns for a particular class. One can simply upsample the channel assigned to a particular class.
>
> >One major weakness is that the method is only compared with the basic resnet and densenet models
>
> The main intended selling points are the following:
> * Introduce neuron dependency/expressivity as two factors of overfitting
> * Show that exchanging only output layers without touching the encoder can improve on these factors and act as strong regularizers without being complex in terms of hyperparameter search and implementation.
>
> We included datasets that are prone to overfitting, and mainly selected architectures that we thought were strong enough to overfit in these settings to have a good way to compare between layers. It is likely that stronger models would lead to even worse overfitting in the baseline. For example, in Table 9 of the appendix, we see that a ResNet with 3 blocks instead of 4 (i.e. a simpler net) leads to better performance in small datasets.
> We did not combine with other regularization methods such as dropout/-connect because we intended to compare the proposed output layers with existing regularizers. However, we use weight decay and common data augmentation (A.1 appendix).
>
> >In table 2, IN-top1 is only 76.36% for baseline. However, it seems that the Resnet-50 can achieve about 77.15% as shown in https://paperswithcode.com/sota/image-classification-on-imagenet. Is there some difference?
>
> Yes, we first cite from the ResNet paper: ‘For best results, we adopt the fully-convolutional form..., and average the scores at multiple scales (images are resized such that the shorter side is in {224, 256, 384, 480, 640}).’ [3]
>
> We don’t use this particular type of inference which might lead to slightly better results. Just for completeness, the difference in top-5 accuracy is smaller (93.29% vs. 93.12% (ours), which is not significant). In addition, the same setting is used to compare all output layers as well as other regularizers. More information about our implementation can be found in Sect. A.1 (appendix).
>
> Thank you again for your feedback and help in improving this paper.
>
> ---
> References:
>
> [1] Zhou, Bolei, et al. "Learning deep features for discriminative localization." Proceedings of the IEEE conference on computer vision and pattern recognition. 2016.
>
> [2] Selvaraju, Ramprasaath R., et al. "Grad-cam: Visual explanations from deep networks via gradient-based localization." Proceedings of the IEEE international conference on computer vision. 2017.
>
> [3] He, Kaiming, et al. "Deep residual learning for image recognition." Proceedings of the IEEE conference on computer vision and pattern recognition. 2016.

---

> > ### Comment · Reviewer_kZyP · 2021-08-23
> > **Thanks for the responses**
> >
> > I thank the authors for their responses to my questions.
> > From my understanding, this work is interesting at least in some ways. As for the experiments, e.g., large datasets and large models, this may not be a simple way to best work. I would like to increase my score.

---

### Official Review · Reviewer_oqDg · 2021-07-20

**Rating:** 6
**Confidence:** 3

**Summary:**

This paper designs and studies regularization effects for different types of output layers on training deep neural networks, which is very useful in practice. They propose five output layer designs and conduct various experiments including a case study. The results show that their proposed designs can achieve improvements of up to 10% on evaluation datasets. Overall, this paper is well-written and easy to follow.

**Limitations And Societal Impact:**

This has been discussed in the paper.

**Main Review:**

Pros:
1) The authors propose two important factors, neuron dependency and expressivity, for  measuring the regularization effect on different output layer designs.
2) The authors conduct various experiments and a case study to validate their designs and the results show up to 10% improvements in accuracy for validation datasets.

Cons:
1) In table 1, I wonder why random fixed layers can achieve such high performance in accuracy, even better than baseline. Is there any explanation?
2) It is not very clear to me why Gradient⊙Activation can measure the dependency. The authors should give a brief discussion/explanation here.
3) The novelty of this paper is somewhat limited. The proposed output designs seem very similar to existing approaches, e.g., w_random - random project, w_sparse - dropout.
4)  What is the difference between w_spare and dropout regularization? What is the difference between w_ensemble and simultaneously training multiple classifiers?
5) Figure 2 can be improved as it is a little bit hard to tell the difference by color. I recommend using different groups of colors or different types of line styles.


**Time Spent Reviewing:**

4

---

> ### Author Response · Authors · 2021-08-09
> **Thanks and response to concerns**
>
> Thanks a lot for your detailed review. We provide answers to your main concerns in the following.
>
> >In table 1, I wonder why random fixed layers can achieve such high performance in accuracy, even better than baseline. Is there any explanation?
>
> All settings in Table 1 correspond to small and/or fine-grained classification tasks without pretraining, i.e., settings that are prone to overfitting and would likely benefit from regularization. We show that simple adjustments to standard output layers, including random fixed layers, provide regularization and thus improve performance.
>
> Random fixed layers provide regularization by reducing neuron dependencies and increasing neuron expressivities, which we identified as two factors of overfitting. This relationship is shown, for example, in Fig. 3 (left), where we see a correlation between dataset size and dependency, and in Figs. 7 and 8 in the appendix, which compare dependencies/expressivities in CUB-200 with and without pretraining.
>
> Also note that in Table 2, which compares classification results in large-scale and transfer learning settings, random fixed layers are comparable to the baseline layer because overfitting is less problematic.
>
>
> >It is not very clear to me why Gradient⊙Activation can measure the dependency.
>
> Intuitively speaking, Gradient $\odot$ Activation provides importance scores for each neuron (in the penultimate layer) w.r.t. a given class. For example, in instance-based dependency, we use it to find the most important neuron for a given instance and the output (argmax) class. If this most important neuron is set to 0, the prediction will be affected the most.
>
> Gradient $\odot$ Activation has two components:
> * The activation value of a neuron $a_n$
> * The rate of change of the output class probability when the activation is changed by an infinitesimally small amount
>
> Thus, a neuron is considered important if its activation value is large and the output class probability increases when the neuron's activation is increased.
>
> Note that the softmax value, which depends on all classes, is used in Eq. 1 because an activation can increase the probability of the output class also by decreasing the logits of other classes.
>
> Furthermore, other approaches such as occlusion [1] are suitable to measure neuron dependencies but are more expensive to compute.
>
> >The novelty of this paper is somewhat limited.
>
> In our opinion, the main novelties are:
> * Identification of neuron dependency/expressivity as two factors of overfitting
> * Showing that exchanging only output layers without touching the encoder can improve on these factors and act as strong regularizers without being complex in terms of hyperparameter search and implementation
>
> >What is the difference between w_sparse and dropout regularization?
>
> Dropout sets activations to 0 at random in each iteration and uses all activations during inference. $W^{sparse}$, on the other hand, uses all activations, fixes all connections to their initialized values, and sets some random connections initially to 0. Dropout changes state in each iteration and has different training/testing behavior, which does not hold for $W^{sparse}$.
>
> Empirically, $W^{sparse}$ achieves better classification (regularization) results than dropout, as shown in Table 5. Moreover, $W^{sparse}$ is not prone to hyperparameter choices compared to dropout, which is shown in Sect. 5.7.
>
> We added a discussion about these differences to Sect. 4.4.
>
> >What is the difference between w_ensemble and simultaneously training multiple classifiers?
>
> Simultaneously training multiple classifiers is a viable approach, but expensive in terms of training time, tuning time, required hardware (if run in parallel) and CO2 consumption.
> $W^{ensemble}$ uses a single network and the full capacity of the penultimate layer to form an ensemble. Although it is an ensemble method, it can be somewhat more efficient compared to standard output layers (see Table 6).
>
> > Figure 2 can be improved as it is a little bit hard to tell the difference by color.
>
> True. We adjusted Fig. 2 by coloring arrows (green: large weight, orange: weight value close to 0, red: large negative weight). Same applies for activations. We have bolded the model version in the caption and figure to emphasize that two different models are being discussed.
>
> Thank you again for your feedback.
>
> ---
> References:
>
> [1] Zeiler, Matthew D., and Rob Fergus. "Visualizing and understanding convolutional networks." European conference on computer vision. Springer, Cham, 2014.

---

> > ### Comment · Reviewer_oqDg · 2021-09-02
> > **Post-rebuttal Response**
> >
> > I thank the authors for their detailed response. Most of my concerns have been addressed. I tend to update my rating to 6. However, I think this paper still needs some further improvements to be clear accepted, such as adding more studies on different backbones (also suggested by other reviewers), including some comparisons between their output layer strategies and adding a random projection layer in the output [1].
> >
> > References:
> > [1] Random Projection in Deep Neural Networks, Piotr Iwo Wójcik, 2018.

---

### Author Response · Authors · 2021-08-09
**Thanks for the reviews and key paper changes**

We thank all reviewers for their thoughtful reviews. The reviewers appreciated the analysis of neuron dependency and expressivity (oqDg, kZyP, V64q), the strong regularization performance of proposed output layers (oqDg, kZyP, V64q), and the clear and easy to follow presentation (oqDg, kZyP).

Since the concerns voiced by the reviewers are mainly non-overlapping, we provide detailed individual responses to each review. Key changes made to the paper are summarized in the following:
* Added additional experiments in Sect. 5.4., where (1) dropout is applied to the encoder, and (2) dropout is combined with $W^{sparse}$ and $W^{ensemble}$
* Added experiments on the impact of weight decay in the baseline output layer to the appendix
* Added a discussion about the differences between $W^{sparse}$ and dropout/-connect in Sect. 4.4
* Added a discussion about the differences between $W^{sparse}$, $W^{1to1}$ and $W^{ensemble}$ in Sect. 4.5 and 4.6
* Adjusted Sect. 3.3 to account for the fact that saliency is used to measure neuron dependency/expressivity
* Added technical details to Sect. 4 (i.e. smoothing in $W^{scaled}$, relation of $\alpha$ and number of classes, $\alpha$ as softmax temperature in $W^{ensemble}$)
* Figure changes: added color to Fig. 2, added input neurons to 1-to-1/ensemble layers
* Polished text to fit to changes

---

### Decision · Program_Chairs · 2021-09-27

**Decision:**

Reject

**Comment:**

This work experiments with various strategies for improving addressing overfitting and regularization of the output layer. In particular, the authors explore ideas such as activation scaling, fixed randomization, sparsification, and ensembling as regularization techniques.This paper also introduces two methods, neuron dependency and expressivity, for measuring the degree of overfitting. The authors tested their ideas out on various image classification tasks including STL-10, CUB-200, Cars-196 and Food-101.

The reviewers were interested in the results and the techniques introduced for measuring overfitting. However, the reviewers also identified several concerns about the thoroughness of the experiments, and encouraged the authors to perform additional experiments and ablations to add more evidence for the viability of the methods. Additionally, no reviewer stepped forward to strongly advocate for the acceptance of this work. For all of these reasons, this paper will not be accepted at NeurIPS, but the authors are heavily encouraged to perform additional experiments and submit this to future venues.